# Circulating microRNA Biomarker for Detecting Breast Cancer in High-Risk Benign Breast Tumors

**DOI:** 10.3390/ijms24087553

**Published:** 2023-04-20

**Authors:** Vedbar S. Khadka, Masaki Nasu, Youping Deng, Mayumi Jijiwa

**Affiliations:** Department of Quantitative Health Sciences, John A. Burns School of Medicine, University of Hawaii, 651 Ilalo Street, Honolulu, HI 96813, USA; vedbar@hawaii.edu (V.S.K.); mnasu@hawaii.edu (M.N.)

**Keywords:** breast cancer, benign breast tumor, high-risk benign breast lesion, plasma, miRNA biomarker, proteomics, IGF-1

## Abstract

High-risk benign breast tumors are known to develop breast cancer at high rates. However, it is still controversial whether they should be removed during diagnosis or followed up until cancer development becomes evident. Therefore, this study sought to identify circulating microRNAs (miRNAs) that could serve as detection markers of cancers arising from high-risk benign tumors. Small RNA-seq was performed using plasma samples collected from patients with early-stage breast cancer (CA) and high-risk (HB), moderate-risk (MB), and no-risk (Be) benign breast tumors. Proteomic profiling of CA and HB plasma was performed to investigate the underlying functions of the identified miRNAs. Our findings revealed that four miRNAs, hsa-mir-128-3p, hsa-mir-421, hsa-mir-130b-5p, and hsa-mir-28-5p, were differentially expressed in CA vs. HB and had diagnostic power to discriminate CA from HB with AUC scores greater than 0.7. Enriched pathways based on the target genes of these miRNAs indicated their association with IGF-1. Furthermore, the Ingenuity Pathway Analysis performed on the proteomic data revealed that the IGF-1 signaling pathway was significantly enriched in CA vs. HB. In conclusion, these findings suggest that these miRNAs could potentially serve as biomarkers for detecting early-stage breast cancer from high-risk benign tumors by monitoring IGF signaling-induced malignant transformation.

## 1. Introduction

Due to advancements in breast cancer screening technology and increased awareness of breast cancer screening, the incidence of female invasive breast cancer has been increasing by approximately 0.5% per year since the mid-2000s. Meanwhile, the use of evolving treatment strategies such as neoadjuvant therapy and novel molecularly targeted agents has contributed to a decrease in breast cancer mortality rates [1]. Mammography is the most commonly used noninvasive and relatively inexpensive diagnostic tool for breast cancer screening. If an abnormality is found, a core needle biopsy (CNB) is generally performed for tissue diagnosis unless the lesion is identified on imaging as a typical no-risk benign tumor. For overtly malignant lesions diagnosed by CNB, there are well-defined standard protocols for patient management, including resection and multimodal therapy. However, for high-risk benign lesions, such as atypical ductal hyperplasia (ADH), atypical lobular hyperplasia (ALH), and Lobular Carcinoma In Situ (LCIS), there is still no consensus on optimal patient management despite decades of exhaustive research.

Breast tumors that are histopathologically diagnosed as certain benign breast conditions have varying levels of breast cancer risks. Proliferative lesions without atypia, such as ductal hyperplasia (without atypia), complex fibroadenoma, sclerosing adenosis, papilloma, or radial scar, are considered moderate-risk and double the risk when compared to no-risk benign lesions. Meanwhile, ADH and ALH increase the risk of developing Ductal Carcinoma In Situ (DCIS) and invasive breast cancer by 4–5 folds, making them high-risk lesions [2,3]. Moreover, high-risk benign lesions diagnosed with CNB have a non-negligible rate of coexisting cancer in the resected tissue, called “upgrade”, with reported mean upgrade rates of 23%, 13.4%, and 19.8% for ADH, ALH, and LCIS, respectively [4]. Hence, National Comprehensive Cancer Network guidelines 2021 recommend surgical excision of these lesions upon identification by CNB [5]. However, the absolute risk of breast cancer in women with ALH and ADH is approximately 1–2%/year, and for LCIS, it is 2%/year [6,7,8]. Coopey et al. estimated a 10-year risk of breast cancer of 21% in women with high-risk benign lesions [6], and Hartmann et al. found a cumulative incidence of 30% for invasive breast cancer after 25 years of follow-up in those with these lesions [9,10]. Therefore, surveillance with careful monitoring may be a viable option for patients who wish to delay surgery until significant changes occur in imaging or clinical findings. Several studies have provided evidence for the selection of patients with high-risk benign lesions who can qualify for close clinical observation with imaging [11,12,13,14,15,16]. Indeed, 1–3% of patients refuse immediate surgery despite the high incidence of cancer [17,18]. This underscores the urgent need for more sensitive and specific biomarkers to detect breast cancer in high-risk benign lesions and complement mammography. Currently, there is no blood-based biomarker that can be used to diagnose cancer as well as benign tumors in terms of the cancer risk level. Such biomarkers could provide a new approach to the management of high-risk benign lesions, mitigating the difficulty in managing these conditions.

Tumor-derived miRNAs present in readily accessible blood samples can serve as circulating biomarkers for detecting cancer [19]. These small non-coding RNAs are stable in the bloodstream and play key roles in post-transcriptional gene regulation [20,21]. The dysregulation of miRNA expression can result in the clinical progressions of cancer through various biological processes such as cell proliferation, differentiation, and apoptosis [22]. In recent years, extensive research has been conducted on the use of miRNAs as diagnostic and therapeutic biomarkers in breast cancer liquid biopsies. According to Arun et al., breast cancer subtypes exhibit different dysregulation in miRNA expression [23]. They speculated on the potential use of miRNAs not only as diagnostic and prognostic biomarkers but also as novel targets specific to breast cancer subtypes. It is possible that miRNAs may play a similar role in benign breast tumors.

In this study, we sought to identify miRNA biomarkers that focus on the detection of cancer development in high-risk benign tumors. To achieve this, we profiled miRNAs in plasma samples obtained from patients with breast cancer and varying risk levels of benign tumors. We identified several miRNAs that were differentially expressed and had diagnostic potential in discriminating between early-stage breast cancer and high- or no-risk benign tumors. Additionally, we investigated the association between these miRNAs and circulating proteins using a bioinformatic approach. The results suggested that these miRNAs have the potential to monitor the progression of high-risk benign tumors into cancer induced by the IGF signaling pathway.

## 2. Results

### 2.1. Plasma Sample Characteristics

A total of 36 plasma samples comprising the cohort included the following four breast tumor groups: CA (*n* = 9), HB (*n* = 14), MB (*n* = 4), and Be (*n* = 9) (Appendix A). The age of the subjects ranged from 38 to 73 years, and their BMI ranged from 18 to 39. The mean age of each group was 54.2 to 59.2 years. The mean BMI of each group was 26 to 28.9, which fell into the overweight category for women (BMI: 25–29.9). Both age (*p* = 0.873) and BMI (*p* = 0.415) were not significantly different among the groups. In the cohort, 17 subjects had never smoked, while 15 were former or current smokers. Eight of nine subjects in the CA group were current or former smokers, whereas 10 of 14 in the HB group, one of four in the MB group, and five of nine in the Be group had no smoking history. The CA samples were further classified by pTNM-Stage (Stage 0 = 1, Stage I = 8), Grade (I = 5, III = 3, not available = 1), and Subtype (Luminal A; ER/PR positive, HER2 negative = 5, Triple Negative; ER/PR/HER2 negative = 4) (Appendix A).

### 2.2. Profiling of Differentially Expressed miRNAs in the Plasma of CA, HB, MB vs. Be

The mapping statistics of several small RNAs were explored, among which miRNAs from miRBase (https://www.mirbase.org (accessed on 15 January 2023)) were mostly detected at an average of 88–92% variation in all four groups. In addition to miRNAs, the distribution of several RNA types was detected, including rRNA, snoRNA (small nucleolar RNA), snRNA (small nuclear RNA), tRNA, vRNA (vault RNA), yRNA, and a few percentages of other RNAs and unassigned reads (Figure 1A). We first performed miRNA expression profiling on CA vs. Be, HB vs. Be, and MB vs. Be and identified 76 differentially expressed miRNAs (Figure 1B). Among the 76 miRNAs, 10, 6, and 51 miRNAs were specific to CA, HB, or MB vs. Be, respectively, while three were common in CA, HB, and MB vs. Be (Figure 1B). The expression patterns of these 76 miRNAs were presented in a heatmap (Figure 1C). Nine out of the 76 miRNAs, namely, hsa-mir-18a-5p, hsa-mir-20a-5p, hsa-mir-99a-5p, hsa-mir-141-3p, hsa-mir-200a-3p, hsa-mir-215-5p, hsa-mir-361-3p, hsa-mir-362-5p, and hsa-mir-3613-5p, were differently expressed in at least two comparisons (Table 1). The three miRNAs common in all comparisons were hsa-mir-141-3p, hsa-mir-200a-3p, and hsa-mir-215-5p, which were all down-regulated in CA, HB, or MB with respect to Be (Table 1, Appendix A). These three miRNAs were found to be linked with breast cancer as per miRNA-disease target association using miRNet (Figure 1D). 

### 2.3. Profiling of Differentially Expressed miRNAs in the Plasma of CA vs. HB

To identify miRNAs that can detect cancer in high-risk benign breast tumors, we profiled miRNAs that were differentially expressed between CA and HB. A total of 15 miRNAs were identified as differentially expressed in CA vs. HB, among which six were up- and nine were down-regulated in CA with respect to HB (Table 2, Figure 2A, Appendix A). Of the 15 miRNAs, only 10 appeared to coincide in the list of 76 differentially expressed miRNAs that were significant against no-risk benign as control. The target genes of these 15 miRNAs were further explored from the proteomic study. 

### 2.4. Plasma Proteome Profiling in Breast Cancer and High-Risk Benign Tumors

To detect the proteins involved in cancer development in high-risk benign breast tumor, CA and HB samples were subjected to proteomics. Proteomics data analysis in CA and HB detected 422 proteins, and seven proteins were differentially expressed in CA vs. HB. Among the seven proteins, two proteins (MMRN1, CORO1A) were up- and five proteins (PZP, SLC4A1, LAMB1, NAGLU, PRDX1) were down-regulated in CA vs. HB (Figure 2B). Moreover, out of the 422 expressed proteins, 72 were identified as targets of 15 differentially expressed miRNAs in CA vs. HB based on miRPath. Spearman correlation between 72 target proteins and 15 differentially expressed miRNAs resulted in 39 proteins having at least one significant association with one of the miRNAs (Appendix A). In CA, PZP, KRT2, C1RL, LAMB1, and NRP1 were significantly associated with three or more miRNAs, while in HB, FLNA, SPTB, TGFBI, ITGB3, LTF, PKM, and ACTBL2 were significantly associated with three or more miRNAs. FLNA and LTF were significantly correlated only in HB. IGF-1 and IGFBP3 were among the 39 proteins. Heatmaps of correlation coefficients for the 39 proteins and 15 miRNAs in CA and HB were shown in Figure 2C,D.

### 2.5. Building and Evaluating Diagnostics Models 

To determine the diagnostic efficacy of differentially expressed miRNAs, several models were built using logistic regression with 10-fold cross-validation on log2 transformed DESeq2 normalized expression data. The three miRNAs (hsa-mir-215-5p, hsa-mir-200a-3p, and hsa-mir-141-3p) common in all comparisons against Be, both individually or as a pair, had at least 70% diagnostic power to discriminate CA from Be. Additionally, among the 15 differentially expressed miRNAs in CA vs. HB, four miRNAs (hsa-mir-128-3p, hsa-mir-130b-5p, hsa-mir-28-5p, and hsa-mir-421) and their pairs showed above 70% diagnostic power to discriminate CA from HB. All potential miRNAs with their performance parameter such as TP Rate, FP Rate, Precision, Recall, F-Measure, and AUC to discriminate early-stage cancer either from no-risk or high-risk were measured (Table 3). 

### 2.6. miRNA Functional Analysis

The four potential miRNA biomarkers (hsa-mir-28-5p, hsa-mir-128-3p, hsa-mir-130b-5p, and hsa-mir-421) that were able to discriminate CA from HB were subjected to pathway and network analyses. These four miRNAs targeted 1162 genes, and the Pathways in Cancer was the most significantly enriched KEGG pathway (adj *p*-value < 0.001) with 45 target genes (Figure 3A). IGF-1, which is known to promote cancer cell growth and proliferation, was targeted by all four miRNAs in the miRNA-target network (Figure 3A). Reactome analysis was conducted to determine the roles of the target genes. The top 10 statistically significant (adj *p*-value < 0.01) groups included Cellular responses to stress (40 genes), Signaling by ERBB4 (27 genes), Gene Expression (95 genes), Diseases of signal transduction (36 genes), Signaling by PDGF (29 genes), Signaling by SCF-KIT (24 genes), Downstream signal transduction (26 genes), Disease (74 genes), Signaling by ERBB2 (25 genes), and Cellular Senescence (24 genes) (Figure 3B). Interestingly, IPA within the proteomic study also showed that the IGF-1 signaling pathway was most significantly enriched in CA vs. HB (Figure 3C), which was consistent with the results of miRNA network analysis.

## 3. Discussion

The incidence of breast cancer in high-risk benign breast tumors is approximately four to five times higher than that in no-risk benign tumors [24]. However, despite decades of ongoing investigative efforts, there is still no clear consensus on whether immediate surgery or imaging surveillance should be considered during diagnosis [25]. Given this situation, we aimed to identify plasma-based biomarkers to serve as a companion tool for optimal treatment selection. Specifically, this study sought to identify biomarkers (1) to examine the presence of cancer when high-risk benign breast tumors are diagnosed and (2) to detect the onset of cancer as early as possible during surveillance if surgery is not performed. 

In this study, plasma miRNAs from female Caucasian patients with early-stage breast cancer and high- to no-risk benign tumors were profiled. Among the miRNAs that were differentially expressed in tumors with malignant potential compared to those without risk, we identified three miRNAs: hsa-mir-215-5p, hsa-mir-200a-3p, and hsa-mir-141-3p. The miRNA-disease target network analysis linked these miRNAs to breast cancer, and they were all down-regulated in CA, HB, and MB when compared to Be. These findings suggested their potential role in the antineoplastic transformation of tumor cells. Moreover, they had a diagnostic power above 70% (AUC > 0.7) to discriminate early cancer from no-risk benign.

In support of our findings, several studies have shown that hsa-miR-215-5p acts as a tumor suppressor in certain cancers. Gao et al. reported that hsa-miR-215-5p suppresses the aggressiveness of breast cancer cells by targeting Sox9 and is down-regulated in breast cancer with respect to normal tissues [26]. In colorectal cancer (CRC), Cheng et al. reported that hsa-mir-215-5p is bound to E2F1/3, resulting in cell cycle arrest during the G0/G1 phase [27]. Wang et al. suggested that low miR-215 expression was significantly associated with high TRAF5 expression, leading to the development of CRC by affecting the tumor microenvironment, including the cytokine level [28].

Notably, both hsa-mir-200a-3p and hsa-mir-141-3p are two of the five members of the miR-200 family (miR-200a, miR-200b, miR-200c, miR-141, and miR-429) sharing the same seed sequence “AACACUG” [29]. Several studies have been conducted on circulating miR-200 family transcripts as diagnostic and prognostic biomarkers of breast cancer in both tissue and serum. In a review from Fontana et al., the miR-200 family was consistently found to be down-regulated in the tissues of more aggressive molecular variants of breast cancer, which is indicative of its well-known function as a repressor of epithelial–mesenchymal transition (EMT) events, an important mechanism by which tumor cells can exhibit more mobile phenotypes to metastasize [30]. However, in contrast to tissue expression patterns, the behavior of circulating miR-200 family genes is inconsistent. Several studies have shown that the miR-200 family is up-regulated in the serum of patients with breast cancer metastasis compared to those without [31,32]. In contrast, Wu et al. reported that serum miR-200a-3p was down-regulated in patients with stage I or II cancer compared to healthy controls [33], and Antolin et al. reported that the levels of miR-141-3p in unfractionated blood samples did not differ significantly between cancer and control groups [34].

Among 15 miRNAs that were differentially expressed in CA vs. HB, four had diagnostic power above 70% (AUC > 0.7) to discriminate early-stage cancer from high-risk benign breast tumors. These miRNAs, hsa-mir-128-3p and hsa-mir-421, were down-regulated, whereas hsa-mir-130b-5p and hsa-mir-28-5p were up-regulated in CA. In several studies, hsa-mir-128-3p was reported to be involved in EMT through the TGF-β1 signaling pathway and in the regulation of cancer stem cells through the Wnt signaling pathway [35,36]. Preclinical studies have demonstrated that the down-regulation of miR-128-3p promotes breast cancer metastasis and maintains stemness [35,37]. As a tumor suppressor, miR-128-3p halts cells at the G0/G1 phase by affecting the levels of cyclin-dependent kinases (CDKs) and cyclin proteins [38]. The expression level of miR-128-3p was lower in breast cancer tissues than in adjacent normal tissues [35] and was much lower in the plasma samples from breast cancer patients than from fibroma patients and healthy controls [37]. These results provide a mechanistic explanation of the possible importance of mir-128-3p as a biomarker for the malignant transformation of benign tumors.

Hsa-miR-421 has been reported to exhibit both carcinogenic and tumor-suppressive properties in many types of cancers. Pan et al. reported the down-regulation of miR-421 in breast cancer tissues and metastatic cell lines [39]. They demonstrated that miR-421 suppresses breast cancer metastasis by directly inhibiting the expression of Metastasis Associated 1 (MTA1). In contrast, Hu et al. reported a higher expression of miR-421 in breast cancer tissues than in adjacent non-tumor tissues and found that it promotes cell proliferation and colony formation in vitro [40]. Similarly, Wang et al. found significant up-regulation of miR-421 in breast cancer tissues and suggested that it promotes breast cancer cell proliferation and migration by suppressing the expression of Programmed Cell Death 4 (PDCD4) [41]. Chen et al. examined plasma mir-421 levels in patients with gastric cancer or precancerous lesions using an experimental design that closely resembled ours [42]. They found that mir-421 levels were significantly higher in the plasma of Stage I cancer patients than in that of precancerous patients. Although their results contradicted ours, it demonstrates both the potential of miR-421 in plasma to detect early-stage tumors and the importance of cellular context when considering new liquid biopsy candidates.

In breast and gastric cancer, hsa-mir-130b-5p has been reported to promote tumorigenesis [43,44]. Miao et al. found it to be up-regulated in breast cancer tissues compared to adjacent tissues, which was consistent with our study in plasma. They observed that it inversely regulated PTEN expression and promoted tumor growth in a mouse xenograft model [43]. Wang et al. reported significantly higher serum hsa-mir-130b-5p levels in early-stage breast cancer patients than in control groups, including benign breast diseases and healthy individuals [45]. They found that serum hsa-mir-130b-5p levels decreased after tumor resection. Zhang et al. reported similar results, showing higher serum hsa-mir-130b-5p levels in patients with hepatocellular carcinoma compared to healthy controls, and the levels decreased after surgery [46].

Products of the miR-28 family can act as tumor suppressors or oncogenes in a variety of malignancies, even in the same type of malignancy, by modulating gene expression and the downstream signaling network [47]. In breast cancer, only tumor suppressive effects have been reported to date [48,49,50]. Circulating levels of hsa-mir-28-5p have not been reported in breast cancer but have been confirmed to be down-regulated in lung, prostate, and renal cell carcinoma [51,52,53]. McDonald et al. reported that the plasma mir-28-5p level was significantly lower in high-grade prostate cancer patients, distinguishing them from low-grade cases [53]. In our study, we found mir-28-5p to be up-regulated in breast cancer plasma when compared to high-risk benign plasma and may represent its possible role as an oncogene in the circulatory context.

Network analysis revealed that these four miRNAs target IGF-1. Remarkably, the IPA analysis from the proteomic study using the same plasma samples also identified IGF-1 signaling as the top canonical pathway enriched in CA vs. HB. Several studies have reported that high levels of blood IGFs are associated with an increased risk of breast cancer [54,55,56,57,58,59]. IGF-binding protein (IGFBP) is one of the key regulators of IGF signaling as it functions as a reservoir of serum IGFs [54,55,56,57,59,60]. Based on these studies, increased levels of circulating IGF-1 and IGFBP3 are considered risk factors for breast cancer. In the clinical trial led by Dr. Carol Fabian (ClinicalTrials.gov Identifier: NCT00291096), serum IGF-1/IGFBP3 levels were explored to determine the relative predictive value of the established risk biomarkers for the development of DCIS and/or invasive cancer. In our study, correlation analysis on CA and HB found that the four miRNAs were highly correlated with IGF-1, IGFBP3, and IGF2R, a negative regulator of IGF signaling. Therefore, we anticipated that the four miRNAs would play an important role in the development of cancer in high-risk benign tumors via the IGF axis. This finding further supports the reliability of these miRNAs as biomarkers for detecting cancer in high-risk benign tumors.

miRNAs are regulated by different genes and signaling pathways and are affected by tumor subtypes and the state of the disease at the time. When multiple miRNAs are combined, their effects may become more complex and difficult to predict than the effects of a single miRNA. This can lead to a decrease in the overall predictive power of the model. Additionally, adding features may lead to a drop in AUC scores in a logistic regression model due to overfitting.

Both smoking and overweight are known risk factors for breast cancer [61,62]. In our cohort, the mean BMI for all groups fell into the overweight range, and there were no significant differences between groups. Due to the small sample size, we did not analyze the association between BMI and miRNA expression. As for smoking, 89% of CA had a smoking history, compared to 21% of HB, 25% of MB, and 56% of Be. No clear correlation was found between cancer risk and the smoking rate.

In our study, we recognize that the small sample size is one of the main limitations. Additionally, our study lacked a validation patient cohort. Despite these limitations, our study identified useful miRNA biomarkers that could differentiate early-stage cancer from no-risk or high-risk benign tumors. To reduce false positives, we only considered miRNA markers selected from at least two statistical analyses, and several previous studies further confirmed their important roles in tumorigenesis. Moreover, we found an association between these biomarkers and IGF-1/IGFBP3, which enhances their credibility as biomarkers for cancer detection in high-risk tumors. Nevertheless, we believe that these biomarkers need to be further evaluated on a larger number of plasma samples to validate their use in clinical practice.

In summary, we identified circulating hsa-mir-128-3p, hsa-mir-421, hsa-mir-130b-5p, and hsa-mir-28-5p as potential biomarkers to detect cancer developed in high-risk benign breast tumors. They could be developed into reliable liquid biopsy biomarkers and companion tools for screening by mammography, MRI, and ultrasound to determine cancer risk.

## 4. Materials and Methods

### 4.1. Plasma Sample Preparation

A total of 36 plasma samples collected from Caucasian females were obtained from the Corewell Health Biorepository (Grand Rapids, MI, USA) on behalf of Accio BiobankOnline (Suffolk, UK). These samples included 9 CA, 14 HB, 4 MB, and 9 Be. All plasma samples were purified from blood collected before surgery or systemic treatment. The blood was collected in a vacutainer tube containing EDTA anticoagulant and immediately transported to the laboratory. The tube was centrifuged at 3500 rpm at 23 °C for 10 min. The separated plasma was transferred to cryovials. The cryovials were placed in a freezer puck and frozen at −80 °C overnight. The cryovials were then removed from the freezer puck and stored at −80 °C.

### 4.2. Total RNA Isolation from Plasma

The total RNA was purified from 200 µL of plasma using the miRNeasy Serum/Plasma Kit (QIAGEN). The RNA was eluted with 14 µL of nuclease-free water, and 1 µL of RNase inhibitor was added. The RNA quality and concentration were determined by using the Agilent 2100 Bioanalyzer.

### 4.3. Small RNA-seq Library Preparation

Library preparation was performed by the Genomics and Bioinformatics Shared Resource (GBSR) at the University of Hawaii Cancer Center (UHCC). The QIAseq miRNA Library Kits (QIAGEN) and the QIAseq miRNA NGS 12 Index IL (QIAGEN) were used following the manufacturer’s instructions. The quality of the libraries was validated by the Agilent 2100 Bioanalyzer using a high-sensitivity DNA chip. Finally, the generated libraries were sequenced using the Illumina NextSeq 500 at GBSR.

### 4.4. Analysis of Small RNA-seq Results of Breast Cancer and Benign Breast Tumors

Illumina’s single-end 76 bp reads were explored using FASTQC and then analyzed with the sRNAbench web tool, which is incorporated in the sRNAtoolbox [63]. Within the tool, the raw reads were processed by trimming adapters and UMIs and filtering out those with mean quality Phred scores below 20. The high-quality reads were aligned against GRCh38_p13 human reference genome with MirGeneDB 2.1 as miRNA annotation database using aligner Bowtie with a seed length of 20 with no mismatches. The tool can also profile other small RNAs using RNAcentral release 20 as an annotation database. The mature miRNA read counts (mult. map. adj.) for each sample were combined, and differential expression analyses were performed between different groups (CA vs. Be, HB vs. Be, MB vs. Be, and CA vs. HB) using the DESeq2 R package [64]. A *p*-value < 0.05 and log2FC > |0.5| were considered differentially expressed. Log2 transformed DESeq2 normalized expression data were used for correlation and downstream statistical analyses. Target gene prediction for differentially expressed miRNAs (CA vs. HB) was conducted using miRPath (https://dianalab.e-ce.uth.gr/html/mirpathv3/ (accessed on 17 January 2023)). 

### 4.5. Proteomics in Breast Cancer and High-Risk Benign Breast Tumors

We received a grant for data-independent acquisition (DIA) mass spectrometry for 20 samples from the IDeA National Quantitative Proteomics Resource. Of the 36 plasma samples used for small RNA-seq, we selected 20 samples, including 9 CA and 11 HB. Abundant plasma proteins were depleted with HighSelect Top14 resin (Thermo) according to the manufacturer’s instructions. Proteins were reduced and alkylated prior to digestion with sequencing-grade modified porcine trypsin (Promega) using S-Trap columns (ProtiFi). Tryptic peptides were then separated by reverse-phase XSelect CSH C18 2.5 um resin (Waters) on an in-line 150 mm × 0.075 mm column using an UltiMate 3000 RSLCnano system (Thermo). Peptides were eluted using a 60 min gradient from 98:2 to 65:35 buffer A:B ratio. Buffers A and B were composed of 0.1% formic acid and 0.5% and 99.9% acetonitrile, respectively. The eluted peptides were ionized by electrospray (2.4 kV), followed by mass spectrometric analysis on an Orbitrap Exploris 480 mass spectrometer (Thermo). To assemble a chromatogram library, six gas-phase fractions were acquired on the Orbitrap Exploris with 4 m/z DIA spectra (4 m/z precursor isolation windows at 30,000 resolution, normalized AGC target 100%, maximum inject time 66 ms) using a staggered window pattern from narrow mass ranges using optimized window placements. Precursor spectra were acquired after each DIA duty cycle, spanning the m/z range of the gas-phase fraction (i.e., 496–602 m/z, 60,000 resolution, normalized AGC target 100%, maximum injection time 50 ms). For wide-window acquisitions, the Orbitrap Exploris was configured to acquire a precursor scan (385–1015 m/z, 60,000 resolution, normalized AGC target 100%, maximum injection time 50 ms) followed by 50× 12 m/z DIA spectra (12 m/z precursor isolation windows at 15,000 resolution, normalized AGC target 100%, maximum injection time 33 ms) using a staggered window pattern with optimized window placements. Precursor spectra were acquired after each DIA duty cycle. 

### 4.6. Proteomic Profiling

Following data acquisition, proteomic analysis was conducted at the IDeA National Resource for Quantitative Proteomics at the University of Arkansas for Medical Sciences. Data were searched using an empirically corrected library, and quantitative analysis was performed to obtain a comprehensive proteomic profile. Proteins were identified and quantified using EncyclopeDIA [65] and visualized with Scaffold DIA using 1% false discovery thresholds at both the protein and peptide levels. Protein-exclusive MS2 intensity values were assessed for quality using proteiNorm [66]. The data were normalized using cyclic loess [67], and statistical analysis was performed using linear models for microarray data (limma) with empirical Bayes (eBayes) smoothing to standard errors [67]. For downstream analysis, proteins with *p*-value < 0.05 and log2FC > |0.5| were considered to be differentially expressed. Normalized protein data were used for correlation analysis, and differentially expressed results were subjected to Ingenuity Pathway Analysis (IPA; QIAGEN Inc, https://www.qiagenbioinformatics.com/products/ingenuity-pathway-analysis (accessed on 20 January 2023)). 

### 4.7. Statistical Analysis

ANOVA was used to compare the mean age and body mass index (BMI) among patients from different groups. Spearman correlation analysis was performed between the normalized miRNAs and their target genes expressed in a proteomic experiment. To determine the diagnostic efficacy of differentially expressed miRNAs, several models were built. To avoid over-fitting, logistic regressions with 10-fold cross-validation were used, and performance parameters such as True Positive (TP) Rate, False Positive (FP) Rate, Precision, Recall, F-Measure and Area under the ROC Curve (AUC) were measured using WEKA 3.8. Significant miRNAs with AUC score > 0.7 were used in network and reactome analyses using the miRNet software [68].

## Figures and Tables

**Figure 1 ijms-24-07553-f001:**
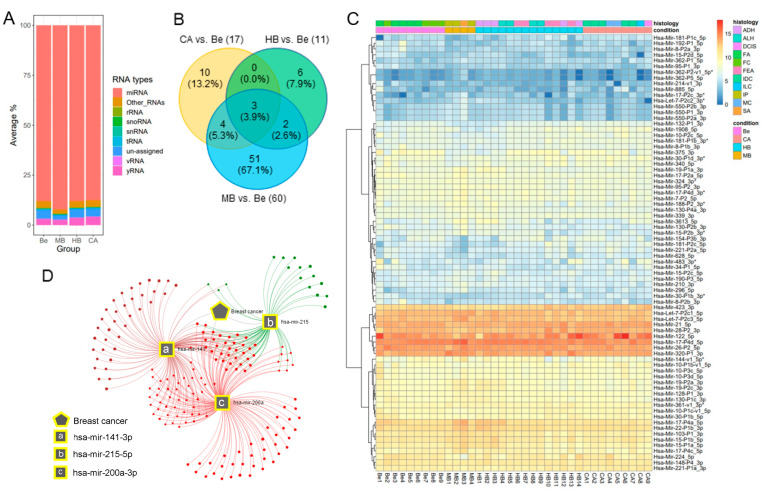
Plasma small RNA profiling. (**A**) Stacked bar plot shows the distribution of RNA species based on the percentage of mapped reads in each group. (**B**) Venn diagram represents the number of differentially expressed miRNAs in CA vs. Be, HB vs. Be, and MB vs. Be. A total of 76 miRNAs were differentially expressed in at least one comparison, while three of them were common in all. (**C**) Heatmap of the 76 differentially expressed miRNAs. (**D**) The miRNA-disease target network highlighted that three overlapped miRNAs were associated with breast cancer.

**Figure 2 ijms-24-07553-f002:**
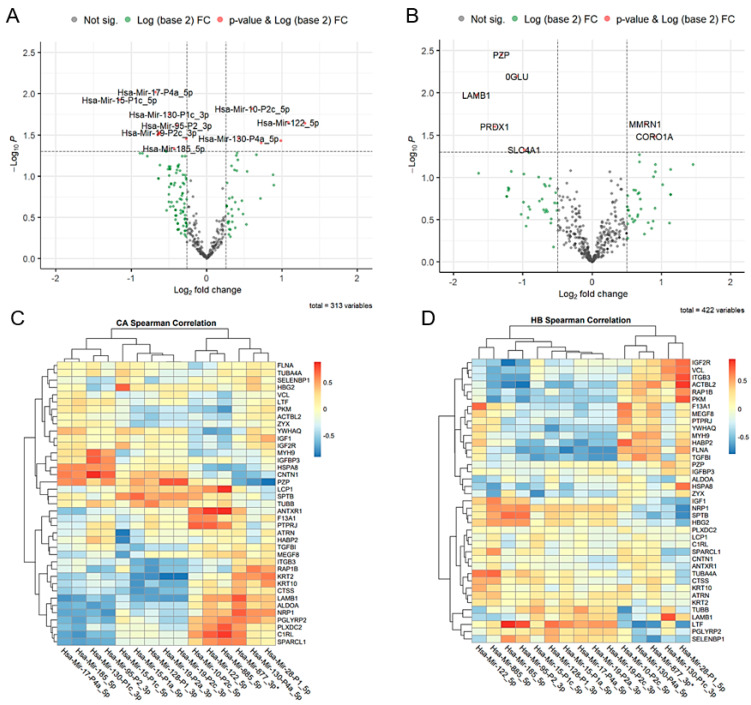
Proteomic data. (**A**,**B**) Volcano plot of differentially expressed miRNAs (**A**) and proteins (**B**) in CA vs. HB. A total of 15 miRNAs and seven proteins were differentially expressed. (**C**,**D**) Heatmap of Spearman correlation coefficients between miRNAs and targeted proteins in CA (**C**) and HB (**D**).

**Figure 3 ijms-24-07553-f003:**
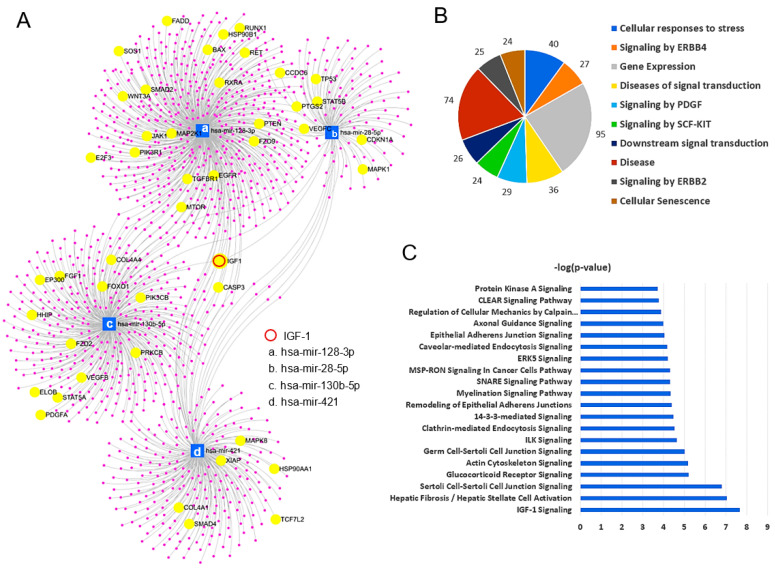
Functional Analysis. (**A**) Network diagram associated with target genes of four miRNAs. Highlighted were 45 target genes associated with Pathways in cancer. (**B**) Roles of top 10 target genes detected from the network analysis. (**C**) Canonical pathways significantly enriched in early-stage cancer with respect to high-risk benign from the proteomic study.

**Table 1 ijms-24-07553-t001:** Differentially expressed miRNAs in multiple comparisons of CA, HB, MB vs. Be.

miRBase ID	MirGeneDB ID	CA vs. Be	HB vs. Be	MB vs. Be
hsa-mir-18a-5p	Hsa-Mir-17-P2a_5p	Down		Up
hsa-mir-20a-5p	Hsa-Mir-17-P4a_5p	Down		Up
hsa-mir-99a-5p	Hsa-Mir-10-P2c_5p	Up		Down
**hsa-mir-141-3p**	**Hsa-Mir-8-P2b_3p**	**Down**	**Down**	**Down**
**hsa-mir-200a-3p**	**Hsa-Mir-8-P2a_3p**	**Down**	**Down**	**Down**
**hsa-mir-215-5p**	**Hsa-Mir-192-P1_5p**	**Down**	**Down**	**Down**
hsa-mir-361-3p	Hsa-Mir-361-v1_3p*		Up	Up
hsa-mir-362-5p	Hsa-Mir-362-P1_5p	Up		Up
hsa-mir-3613-5p	Hsa-Mir-3613_5p		Up	Up

Note: The bold miRNAs were common and down-regulated in CA, HB, MB vs. Be.

**Table 2 ijms-24-07553-t002:** Differentially expressed miRNAs in CA vs. HB.

miRBase ID	MirGeneDB ID	CA vs. HB	¹ Common in 76
hsa-mir-15a-5p	Hsa-Mir-15-P1a_5p	Down	Yes
hsa-mir-19b-3p (19b-1)	Hsa-Mir-19-P2a_3p	Down	Yes
hsa-mir-19b-3p (19b-2)	Hsa-Mir-19-P2c_3p	Down	Yes
hsa-mir-20a-5p	Hsa-Mir-17-P4a_5p	Down	Yes
hsa-mir-28-5p	Hsa-Mir-28-P1_5p	Up	No
hsa-mir-99a-5p	Hsa-Mir-10-P2c_5p	Up	Yes
hsa-mir-122-5p	Hsa-Mir-122_5p	Up	Yes
hsa-mir-128-3p	Hsa-Mir-128-P1_3p	Down	Yes
hsa-mir-130a-3p	Hsa-Mir-130-P1c_3p	Down	Yes
hsa-mir-130b-5p	Hsa-Mir-130-P4a_5p	Up	No
hsa-mir-185-5p	Hsa-Mir-185_5p	Down	No
hsa-mir-421	Hsa-Mir-95-P2_3p	Down	Yes
hsa-mir-424-5p	Hsa-Mir-15-P1c_5p	Down	No
hsa-mir-877-3p	Hsa-Mir-877_3p*	Up	No
hsa-mir-885-5p	Hsa-Mir-885_5p	Up	Yes

Note: ¹ Common in 76 miRNAs represents those significantly different against no-risk benign as control.

**Table 3 ijms-24-07553-t003:** Potential miRNA biomarkers to detect cancer in no- and high-risk benign breast tumors.

	miRBase ID	TP Rate	FP Rate	Precision	Recall	F-Measure	AUC
CA vs. Be	hsa-mir-215-5p	0.667	0.222	0.750	0.667	0.706	0.790
	hsa-mir-200a-3p	0.667	0.444	0.600	0.667	0.632	0.716
	hsa-mir-141-3p	0.667	0.333	0.667	0.667	0.667	0.741
	hsa-mir-215-5p + hsa-mir-200a-3p	0.667	0.333	0.667	0.667	0.667	0.778
	hsa-mir-215-5p + hsa-mir-141-3p	0.556	0.222	0.714	0.556	0.625	0.753
CA vs. HB	hsa-mir-128-3p	0.444	0.214	0.571	0.444	0.500	0.722
	hsa-mir-130b-5p	0.667	0.143	0.750	0.667	0.706	0.746
	hsa-mir-28-5p	0.667	0.143	0.750	0.667	0.706	0.841
	hsa-mir-421	0.556	0.286	0.556	0.556	0.556	0.714
	hsa-mir-28-5p + hsa-mir-421	0.444	0.286	0.500	0.444	0.471	0.746
	hsa-mir-130b-5p + hsa-mir-28-5p + hsa-mir-421	0.667	0.214	0.667	0.667	0.667	0.770

## Data Availability

The data generated in this study are publicly available in the Gene Expression Omnibus (GEO) at GSE225117.

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
