# Peer review of "Circulating microRNA Biomarker for Detecting Breast Cancer in High-Risk Benign Breast Tumors"

_ijms, 2023, doi:10.3390/ijms24087553_

Round 1
Reviewer 1 Report
1. The resolution of figures musted be improved.
2. The number of samples is low and limit this manuscript. The authors should discuss it in the Section of discussion.
Author Response
- The resolution of figures must be improved.
We thank you for your feedback to improve our manuscript. As per your suggestion, we have revised Figures 1, 2, and 3 to enhance the legibility of the text and charts.
- The number of samples is low and limits this manuscript. The authors should discuss it in the Section of discussion.
Thank you for your valuable comments. We agree that the sample size is relatively small. We highlighted the part mentioning this issue in Discussion (Line 312, highlighted).
Reviewer 2 Report
In this study, the authors have identified several miRNA biomarkers for identifying breast cancer using RNA sequencing and computational data analysis. The manuscript is well written and the results are satisfying their hypothesis. My comments are here:
- How the four samples of breast cancers such as CA, HB, MB, and Be have been classified and what are existing biomarkers have already reported? This information should discuss in the introduction part
- It is interesting to know why the less predictive power (AUC) is lower for the combination of different miRNAs than a single one. It may due to these miRNAs could be regulated by different genes/pathways. Do authors would like to comment on this
- The prediction of power (AUC) for top miRNAs should be provided by the ROC curve
- Do the authors have any idea about whether the identified miRNAs biomarkers are sharing any sequence or structural similarity? Also, these biomarkers have been used for other cancer types.
- The resolution of figure 1 should be improved
Author Response
- How the four samples of breast cancers such as CA, HB, MB, and Be have been classified and what are existing biomarkers have already reported? This information should discuss in the introduction part.
We thank you for your careful review and comments to improve our manuscript. CA, HB, MB, and Be were classified histopathologically, and benign tumors were further classified by cancer incidence. We highlighted the part mentioning this (Line 44-49, highlighted) and added small change in the text (Line 42, 43). Sample information is provided as a supplementary Table S1, which includes how we classified our samples.
With expert review, it is possible to determine to some degree whether an abnormal mammogram is cancerous or not, and what type of benign tumor it is. Nevertheless, biopsies are required for an accurate diagnosis (Line 34-36, highlighted). There are no blood biomarkers that distinguish CA, HB, MB, or Be that can be used clinically. We have added this information in the Introduction (Introduction, Line 65, 66).
Some of our identified miRNA biomarkers in plasma samples have been reported in various studies. We have discussed it in the Discussion.
- It is interesting to know why the less predictive power (AUC) is lower for the combination of different miRNAs than a single one. It may due to these miRNAs could be regulated by different genes/pathways. Do authors would like to comment on this.
Thank you for your question and comment regarding AUC scores. As you suggested, these miRNAs could be regulated by different genes/pathways, which are also influenced by cancer and benign tumor subtypes and their conditions. We agree that when multiple miRNAs are combined, their effects may become more complex and difficult to predict than the effects of a single miRNA. This can lead to a decrease in the overall predictive power of the model. Also, adding features may lead to a drop in AUC scores in a logistic regression model due to overfitting. Our goal is to find a balance between the number of features and the model’s ability to generalize to new data. Therefore, we only selected models that had AUC > 0.7. We added our thoughts in the Discussion (Discussion, Line 300-305).
- The prediction of power (AUC) for top miRNAs should be provided by the ROC curve.
Thank you. We have included AUC plots as a Supplementary Figure S1.
- Do the authors have any idea about whether the identified miRNAs biomarkers are sharing any sequence or structural similarity? Also, these biomarkers have been used for other cancer types.
The miRNAs we identified as biomarkers were single-stranded RNAs of 21-23 nt in length and did not show any characteristic secondary structure. hsa-mir-200a-3p and hsa-mir-141-3p are part of the miR-200 family and have a common seed sequence, but no obvious similarities were observed among the other miRNAs. We added the seed sequences and the entire sequences of the 7 miRNAs in the Supplementary Table S4.
As we discussed in the Discussion, all of the miRNAs we identified as biomarkers have been reported to be associated with various types of cancer, but have not yet been applied as biomarkers clinically.
- The resolution of figure 1 should be improved.
We thank you for your feedback to improve our manuscript. We have revised Figure 1 as per your suggestion.
Reviewer 3 Report
This study was based on 36 clinical samples collected by auothors. Briefly, authors performed a Proteomic profiling of CA and HB plasma to investigate the underlying 15 functions of the identified miRNAs. They claimed that four miRNAs, hsa-mir-128-3p, hsa-16 mir-421, hsa-mir-130b-5p, and hsa-mir-28-5p were differentially expressed in CA vs. HB and had 17 diagnostic power to discriminate CA from HB with AUC scores greater than 0.7. Enriched pathways 18 based on the target genes of these miRNAs indicated their association with IGF-1. Furthermore, the 19 Ingenuity Pathway Analysis performed on the proteomic data revealed the IGF-1 signaling pathway 20 was significantly enriched in CA vs. HB. Authors finally concluded that these findings suggest that these miRNAs could potentially serve as biomarkers for detecting early-stage breast cancer from high-risk benign tumors by monitoring IGF signaling-induced malignant transformation. However, apparent flaws exist in the design of the study, therefore serious revision should be performed before acceptance.
1.As we all know, there are different subtypes of breast cancer: 1.luminal A, 2.luminal B,3.human epidermal growth factor receptor 2 (HER2)-enriched, and 4.basal-like. More importantly, breast cancer subtype-specific cellular effects are influenced by distinct miRNAs and a comprehensive network of subtype-specific miRNAs and mRNAs would allow us to better understand breast cancer signaling. (10.3390/biomedicines10030651). Therefore, the miRNAs selected in this manuscript should be analyzed based on certain subtype of breast cancer, not as a whole one (CA). Current results retrieved from data of unclassified breast cancer hard to provide precise conclusion.
2.If in vivo or in vitro experiments can not be added to confirm authors’ hypothesis, large-scale clinical data of public database such TCGA and GWAS, should be applied to confirm their conclusion.
Author Response
1.As we all know, there are different subtypes of breast cancer: 1.luminal A, 2.luminal B, 3.human epidermal growth factor receptor cancer subtype-specific cellular effects are influenced by distinct miRNAs and a comprehensive network of subtype-specific miRNAs and mRNAs would allow us to better understand breast cancer signaling. (10.3390/biomedicines10030651). Therefore, the miRNAs selected in this manuscript should be analyzed based on certain subtype of breast cancer, not as a whole one (CA). Current results retrieved from data of unclassified breast cancer hard to provide precise conclusion.
Thank you for your important suggestions. We agree that a subtype-based approach is fundamental to breast cancer research. In addition, the review article you referred to us was a concise and informative source of research findings on breast cancer subtype-specific miRNAs.
Regarding the miRNA biomarkers we identified in our study, our aim was to develop markers that could detect all subtypes of cancer that develop in high-risk benign tumors. Therefore, we focused on examining the sensitivity of the identified miRNAs, regardless of the cancer subtype. In our sample group, there were 5 Luminal A and 4 Triple Negative cases, and such a small sample size may not allow for a highly accurate statistical examination.
We explored our miRNA biomarkers based on the review article by Arun et al. In the review, down-regulation of miR-200 family was highlighted in TNBC leading to migratory phenotype. Interestingly, in our study, miR-200a-3p and miR-141-3p were down-regulated not only in cancer but also in high-risk and moderate-risk benign tumors. The ability of EMT may be associated with malignant transformation. We have cited this review article in the Introduction to emphasize the potential roles of miRNA in breast cancer and breast benign tumors (Introduction, Line 75-79).
2.If in vivo or in vitro experiments can not be added to confirm authors’ hypothesis, large-scale clinical data of public database such TCGA and GWAS, should be applied to confirm their conclusion.
Thank you for your suggestion to improve our manuscript. We explored available datasets from TCGA and GEO. Due to the lack of suitable plasma datasets on high-risk benign breast tumors in both databases, we were unable to evaluate our miRNA using existing datasets. Instead, we found the following supportive reports.
TCGA had no data on high-risk benign breast tumors, but we did find a small RNA-seq dataset of breast cancer and adjacent normal tissue by Troester et al. They reported that miR-200 family genes were increased in normal tissue compared to cancer tissue, which is consistent with our results from plasma. GEO also had no studies using plasma from patients with benign breast tumors; however, they did have a Tissue Microarray dataset posted by Poola et al. on ADH with and without a history of breast cancer. They found elevated MMP-1 protein level in ADH with a history of cancer. They concluded that MMP-1 degrades IGFBP2 and 3, FGFBP, and TGF-βBP, releasing IGF, FGF, and TGF-β, leading to cancer development. This finding is consistent with our conclusion that the miRNAs that distinguish ADH from cancer may monitor the IGF1 axis.
Round 2
Reviewer 3 Report
Authors performed a Proteomic profiling of CA and HB plasma to investigate the underlying 15 functions of the identified miRNAs, and concluded that their findings suggest that these miRNAs could potentially serve as biomarkers for detecting early-stage breast cancer from high-risk benign tumors by monitoring IGF signaling-induced malignant transformation. This study might provide a novel sight for future research of monitoring benign breast tumor clinically. Acceptance is recommended in current form.